# Least Informative Dimensions

**Fabian H. Sinz**
Department for Neuroethology
Eberhard Karls University Tübingen
fabee@epagoge.de

**Anna Stöckl**
Department for Functional Zoology
Lund University, Sweden
Anna.Stockl@biol.lu.se

**Jan Grewe**
Department for Neuroethology
Eberhard Karls University Tübingen
jan.grewe@uni-tuebingen.de

**Jan Benda**
Department for Neuroethology
Eberhard Karls University Tübingen
jan.benda@uni-tuebingen.de

## Abstract

We present a novel non-parametric method for finding a subspace of stimulus features that contains all information about the response of a system. Our method generalizes similar approaches to this problem such as *spike triggered average*, *spike triggered covariance*, or *maximally informative dimensions*. Instead of maximizing the mutual information between features and responses directly, we use integral probability metrics in kernel Hilbert spaces to minimize the information between uninformative features and the combination of informative features and responses. Since estimators of these metrics access the data via kernels, are easy to compute, and exhibit good theoretical convergence properties, our method can easily be generalized to populations of neurons or spike patterns. By using a particular expansion of the mutual information, we can show that the informative features must contain all information if we can make the uninformative features independent of the rest.

## 1 Introduction

An important aspect of deciphering the neural code is to determine those stimulus features populations of sensory neurons are most sensitive to. Approaches to that problem include white noise analysis [2, 14], in particular spike-triggered average [4] or spike-triggered covariance [3, 19], canonical correlation analysis or population receptive fields [12], generalized linear models [18, 15], or maximally informative dimensions [22]. All these techniques have in common that they optimize a statistical dependency measure between stimuli and spike responses over the choice of a linear subspace. The particular algorithms differ in the dimensionality of the subspace they extract (one- vs. multi-dimensional), the statistical measure they use (correlation, likelihood, relative entropy), and whether an extension to population responses is feasible or not. While spike-triggered average uses correlation and is restricted to a single subspace, spike-triggered covariance and canonical correlation analysis can already extract multi-dimensional subspaces but are still restricted to second-order statistics. Maximally informative dimensions is the only technique of the above that can extract *multiple* dimensions that are informative also with respect to *higher-order* statistics. However, an extension to spike patterns or population responses is not straightforward because of the curse of dimensionality. Here we approach the problem from a different perspective and propose an algorithm that can extract a multi-dimensional subspace containing all relevant information about the neural responses $Y$ in terms of Shannon's mutual information (if such a subspace exists). Our method does not commit to a particular parametric model, and can easily be extended to spike patterns or population responses.

In general, the problem of finding the most informative subspace of the stimuli $\boldsymbol{X}$ about the responses $\boldsymbol{Y}$ can be described as finding an orthogonal matrix $Q$ (a basis for $\mathbb{R}^n$) that separates $\boldsymbol{X}$ into informative and non-informative features $(\boldsymbol{U}, \boldsymbol{V})^\top = Q\boldsymbol{X}$. Since $Q$ is orthogonal, the mutual information $I[\boldsymbol{X} : \boldsymbol{Y}]$ between $\boldsymbol{X}$ and $\boldsymbol{Y}$ can be decomposed as [5]

$$
\begin{aligned}
I[\boldsymbol{Y} : \boldsymbol{X}] &= I[\boldsymbol{Y} : \boldsymbol{U}, \boldsymbol{V}] = \mathrm{E}_{\boldsymbol{X}, \boldsymbol{Y}} \left[ \log \frac{p(\boldsymbol{U}, \boldsymbol{V}, \boldsymbol{Y})}{p(\boldsymbol{U}, \boldsymbol{V})\, p(\boldsymbol{Y})} \right] \\
&= I[\boldsymbol{Y} : \boldsymbol{U}] + \mathrm{E}_{\boldsymbol{Y}, \boldsymbol{V}} \left[ \log \frac{p(\boldsymbol{Y}, \boldsymbol{V} \mid \boldsymbol{U})}{p(\boldsymbol{Y} \mid \boldsymbol{U})\, p(\boldsymbol{V} \mid \boldsymbol{U})} \right] \\
&= I[\boldsymbol{Y} : \boldsymbol{U}] + \mathrm{E}_{\boldsymbol{U}} \left[ I[\boldsymbol{Y} \mid \boldsymbol{U} : \boldsymbol{V} \mid \boldsymbol{U}] \right].
\end{aligned}
\tag{1}
$$

Since the two terms on the right hand side of equation (1) are always positive and sum up to the mutual information between $\boldsymbol{Y}$ and $\boldsymbol{X}$, two ways to obtain maximally informative features $\boldsymbol{U}$ about $\boldsymbol{Y}$ would be to either maximize $I[\boldsymbol{Y} : \boldsymbol{U}]$ or to minimize $\mathrm{E}_{\boldsymbol{U}}[I[\boldsymbol{Y}|\boldsymbol{U} : \boldsymbol{V}|\boldsymbol{U}]]$ via the choice of $Q$.

The first possibility is along the lines of maximally informative dimensions [22] and involves direct estimation of the mutual information. The second possibility which avoids direct estimation has been proposed by Fukumizu and colleagues [5, 6] (we discuss both in Section 3). Here, we explore a third possibility, which trades practical advantages against a slightly more restrictive objective. The idea is to obtain maximally informative features $\boldsymbol{U}$ by making $\boldsymbol{V}$ as independent as possible from the combination of $\boldsymbol{U}$ and $\boldsymbol{Y}$. For this reason, we name our approach *least informative dimensions (LID)*. Formally, least informative dimensions tries to minimize the mutual information between the pair $\boldsymbol{Y}, \boldsymbol{U}$ and $\boldsymbol{V}$. Using the chain rule for *multi information* we can write it as (see supplementary material)

$$
I[\boldsymbol{Y}, \boldsymbol{U} : \boldsymbol{V}] = I[\boldsymbol{Y} : \boldsymbol{X}] + I[\boldsymbol{U} : \boldsymbol{V}] - I[\boldsymbol{Y} : \boldsymbol{U}].
\tag{2}
$$

This means that minimizing $I[\boldsymbol{Y}, \boldsymbol{U} : \boldsymbol{V}]$ is equivalent to maximizing $I[\boldsymbol{Y} : \boldsymbol{U}]$ while simultaneously minimizing $I[\boldsymbol{U} : \boldsymbol{V}]$. Note that $I[\boldsymbol{Y}, \boldsymbol{U} : \boldsymbol{V}] = 0$ implies $I[\boldsymbol{U} : \boldsymbol{V}] = 0$. Therefore, if $Q$ can be chosen such that $I[\boldsymbol{Y}, \boldsymbol{U} : \boldsymbol{V}] = 0$ equation (2) reduces to $I[\boldsymbol{Y} : \boldsymbol{X}] = I[\boldsymbol{Y} : \boldsymbol{U}]$, pushing all information about $\boldsymbol{Y}$ into $\boldsymbol{U}$.

Since each new choice of $Q$ requires the estimation of the mutual information between (potentially high-dimensional) variables, direct optimization is hard or unfeasible. For this reason, we resort to another dependency measure which is easier to estimate but shares its minimum with mutual information, that is, it is zero if and only if the mutual information is zero. The objective is to choose $Q$ such that $(\boldsymbol{Y}, \boldsymbol{U})$ and $\boldsymbol{V}$ are independent in that dependency measure. If we can find such a $Q$, then we know that $I[\boldsymbol{Y}, \boldsymbol{U} : \boldsymbol{V}]$ is zero as well, which means that $\boldsymbol{U}$ are the most informative features in terms of the Shannon mutual information. This will allow us to obtain maximally informative features without ever having to estimate a mutual information. The easier estimation procedure comes at the cost of only being able to link the alternative dependency measure to the mutual information if both of them are zero. If there is no $Q$ that achieves this, we will still get informative features in the alternative measure, but it is not clear how informative they are in terms of mutual information.

## 2 Least informative dimensions

This section describes how to efficiently find a $Q$ such that $I[\boldsymbol{Y}, \boldsymbol{U} : \boldsymbol{V}] = 0$ (if such a $Q$ exists). Unless noted otherwise, $(\boldsymbol{U}, \boldsymbol{V})^\top = Q\boldsymbol{X}$ where $\boldsymbol{U}$ denotes the informative and $\boldsymbol{V}$ the uninformative features. The mutual information is a special case of the relative entropy

$$
D_{KL}[p \| q] = \mathrm{E}_{X \sim p} \left[ \frac{\log p(X)}{\log q(X)} \right]
$$

between two distribution $p$ and $q$. While being linked to the rich theoretical background of Shannon information theory, the relative entropy is known to be hard to estimate [25]. Alternatives to relative entropy of increasing practical interest are the *integral probability metrics (IPM),* defined as [25, 17]

$$
\gamma_{\mathcal{F}}(\boldsymbol{X} : \boldsymbol{Z}) = \sup_{f \in \mathcal{F}} |\mathrm{E}_{\boldsymbol{X}}[f(\boldsymbol{X})] - \mathrm{E}_{\boldsymbol{Z}}[f(\boldsymbol{Z})]|.
\tag{3}
$$

Intuitively, the metric in equation (3) searches for a function $f$, which can detect a difference in the distributions of two random variables $\boldsymbol{X}$ and $\boldsymbol{Z}$. If no such witness function can be found, the

distributions must be equal. If $\mathcal{F}$ is chosen to be a sufficiently rich reproducing kernel Hilbert space $\mathcal{H}$ [21], then the supremum in equation (3) can be computed explicitly and the divergence can be computed in closed form [7]. This particular type of IPM is called *maximum mean discrepancy (MMD)* [9, 7, 10].

A kernel $k : \mathcal{X} \times \mathcal{X} \to \mathbb{R}$ is a symmetric function such that the matrix $K_{ij} = k\left(\boldsymbol{x}_i, \boldsymbol{x}_j\right)$ is positive (semi)-definite for every selection of points $\boldsymbol{x}_1, ..., \boldsymbol{x}_m \in \mathcal{X}$ [21]. In that case, the functions $k\left(\cdot, \boldsymbol{x}\right)$ are elements of a reproducing kernel Hilbert space (RKHS) of functions $\mathcal{H}$. This space is endowed with a dot product $\langle \cdot, \cdot \rangle_{\mathcal{H}}$ with the so called reproducing property $\langle k\left(\cdot, \boldsymbol{x}\right), f \rangle_{\mathcal{H}} = f\left(\boldsymbol{x}\right)$ for $f \in \mathcal{H}$. In particular, $\langle k\left(\cdot, \boldsymbol{x}\right), k\left(\cdot, \boldsymbol{x}'\right) \rangle_{\mathcal{H}} = k\left(\boldsymbol{x}, \boldsymbol{x}'\right)$. When setting $\mathcal{F}$ in equation (3) to be the unit ball in $\mathcal{H}$, then the IPM can be computed in closed form as the norm of the difference between the mean functions in $\mathcal{H}$ [7, 10, 8, 26]:

$$
\begin{aligned}
\gamma_{\mathcal{H}}\left(\boldsymbol{X} : \boldsymbol{Z}\right) &= \left\| \mathrm{E}_{\boldsymbol{X}}\left[k\left(\cdot, \boldsymbol{X}\right)\right] - \mathrm{E}_{\boldsymbol{Z}}\left[k\left(\cdot, \boldsymbol{Z}\right)\right] \right\|_{\mathcal{H}} \qquad (4) \\
&= \left( \mathrm{E}_{\boldsymbol{X}, \boldsymbol{X}'}\left[k\left(\boldsymbol{X}, \boldsymbol{X}'\right)\right] - 2\mathrm{E}_{\boldsymbol{X}, \boldsymbol{Z}}\left[k\left(\boldsymbol{X}, \boldsymbol{Z}\right)\right] + \mathrm{E}_{\boldsymbol{Z}, \boldsymbol{Z}'}\left[k\left(\boldsymbol{Z}, \boldsymbol{Z}'\right)\right] \right)^{\frac{1}{2}},
\end{aligned}
$$

where the first equality is derived in [7], and second equality uses the bi-linearity of the dot product and the reproducing property of $k$. Furthermore, $\left(\boldsymbol{X}, \boldsymbol{X}'\right) \sim P_{\boldsymbol{X}} \times P_{\boldsymbol{X}}$ and $\left(\boldsymbol{Z}, \boldsymbol{Z}'\right) \sim P_{\boldsymbol{Z}} \times P_{\boldsymbol{Z}}$ are two independent random variables drawn from the marginal distributions of $\boldsymbol{X}$ and $\boldsymbol{Z}$, respectively.

The function $\mathrm{E}_{\boldsymbol{X}}\left[k\left(\cdot, \boldsymbol{X}\right)\right]$ is an embedding of the distribution of $\boldsymbol{X}$ into the RKHS $\mathcal{H}$ via $\boldsymbol{X} \mapsto \mathrm{E}_{\boldsymbol{X}}\left[k\left(\cdot, \boldsymbol{X}\right)\right]$. If this map is injective, that is, if it uniquely represents the probability distribution of $\boldsymbol{X}$, then equation (4) is zero if and only if the probability distributions of $\boldsymbol{X}$ and $\boldsymbol{X}'$ are the same. Kernels with that property are called *characteristic* in analogy to the characteristic function $\phi_{\boldsymbol{X}}\left(\boldsymbol{t}\right) \mapsto \mathrm{E}_{\boldsymbol{X}}\left[\exp\left(i\boldsymbol{t}^{\top} \boldsymbol{X}\right)\right]$ [26, 27]. This means that for characteristic kernels MMD is zero exactly if the relative entropy $D_{KL}\left[p\|q\right]$ is zero as well. Since the mutual information is the relative entropy between the joint distribution and the products of the marginals, we can use MMD to search for a $Q$ such that $\gamma_{\mathcal{H}}\left(P_{\boldsymbol{Y}, \boldsymbol{U}, \boldsymbol{V}} : P_{\boldsymbol{Y}, \boldsymbol{U}} \times P_{\boldsymbol{V}}\right)$ is zero[1], which then implies that $I\left[\boldsymbol{Y}, \boldsymbol{U} : \boldsymbol{V}\right] = 0$ as well. The finite sample version of (4) is simply given by replacing the expectations with the empirical mean (and possibly some bias correction) [7, 10, 8]. The estimation of $\gamma_{\mathcal{H}}$ therefore only involves summation over three kernel matrices and can be done in a few lines of code. Unlike for the relative entropy, the empirical estimation of MMD is therefore much more feasible. Furthermore, the residual error of the empirical estimator can be shown to decrease on the order of $1/\sqrt{m}$ where $m$ is the number of data points [25]. Note in particular, that this rate does not depend on the dimensionality of the data.

**Objective function**   The objective function for our optimization problem now has the following form: We transform input examples $\boldsymbol{x}_i$ into features $\boldsymbol{u}_i$ and $\boldsymbol{v}_i$ via $\left(\boldsymbol{u}_i, \boldsymbol{v}_i\right) = Q\boldsymbol{x}_i$. Then we use a kernel $k\left(\left(\boldsymbol{u}_i, \boldsymbol{v}_i, \boldsymbol{y}_i\right), \left(\boldsymbol{u}_j, \boldsymbol{v}_j, \boldsymbol{y}_j\right)\right)$ to compute and minimize MMD with respect to the choice of $Q$. In order to do that efficiently, a few adaptations are required. First, without loss of generality, we minimize the squared MMD instead of MMD itself

$$
\gamma_{\mathcal{H}}^2\left(\boldsymbol{Z}_1, \boldsymbol{Z}_2\right) = \mathrm{E}_{\boldsymbol{Z}_1, \boldsymbol{Z}_1'}\left[k\left(\boldsymbol{Z}_1, \boldsymbol{Z}_1'\right)\right] - 2\mathrm{E}_{\boldsymbol{Z}_1, \boldsymbol{Z}_2}\left[k\left(\boldsymbol{Z}_1, \boldsymbol{Z}_2\right)\right] + \mathrm{E}_{\boldsymbol{Z}_2, \boldsymbol{Z}_2'}\left[k\left(\boldsymbol{Z}_2, \boldsymbol{Z}_2'\right)\right], (5)
$$

where $\boldsymbol{Z}_1 = \left(\boldsymbol{Y}, \boldsymbol{U}, \boldsymbol{V}\right) \sim P_{\boldsymbol{Y}, \boldsymbol{U}, \boldsymbol{V}}$ and $\boldsymbol{Z}_2 = \left(\boldsymbol{Y}, \boldsymbol{U}, \boldsymbol{V}\right) \sim P_{\boldsymbol{Y}, \boldsymbol{U}} \times P_{\boldsymbol{V}}$.

Second, in order to get samples from $P_{\boldsymbol{Y}, \boldsymbol{U}} \times P_{\boldsymbol{V}}$, we assume that our kernel takes the form $k\left(\left(\boldsymbol{u}_i, \boldsymbol{v}_i, \boldsymbol{y}_i\right), \left(\boldsymbol{u}_j, \boldsymbol{v}_j, \boldsymbol{y}_j\right)\right) = k_1\left(\left(\boldsymbol{u}_i, \boldsymbol{y}_i\right), \left(\boldsymbol{u}_j, \boldsymbol{y}_j\right)\right) \cdot k_2\left(\boldsymbol{v}_i, \boldsymbol{v}_j\right)$. For this special case, one can incorporate the independence assumption between $\boldsymbol{U}, \boldsymbol{Y}$ and $\boldsymbol{V}$ directly by using the fact that for independent random variables, the expectation of the product is equal to the product of expectations, that is,

$$
\mathrm{E}\left[k_1\left(\left(\boldsymbol{u}_i, \boldsymbol{y}_i\right), \left(\boldsymbol{u}_j, \boldsymbol{y}_j\right)\right) \cdot k_2\left(\boldsymbol{v}_i, \boldsymbol{v}_j\right)\right] = \mathrm{E}\left[k_1\left(\left(\boldsymbol{u}_i, \boldsymbol{y}_i\right), \left(\boldsymbol{u}_j, \boldsymbol{y}_j\right)\right)\right] \mathrm{E}\left[k_2\left(\boldsymbol{v}_i, \boldsymbol{v}_j\right)\right].
$$

This special case of MMD is equivalent to the *Hilbert-Schmidt Independence Criterion (HSIC)* [9, 23] and can be computed as

$$
\hat{\gamma}_{hs}^2 = \frac{1}{\left(m-1\right)^2} \mathrm{tr}\left(K_1 H K_2 H\right), \qquad (6)
$$

where $K_1$ and $K_2$ denote the matrices of pairwise kernel values between the data sets $\left\{\left(\boldsymbol{u}_i, \boldsymbol{y}_i\right)\right\}_{i=1}^m$ and $\left\{\boldsymbol{v}_i\right\}_{i=1}^m$, respectively, and $H_{ij} = \delta_{ij} - m^{-1}$.

Note, however, that one could in principle also optimize (5) for a non-factorizing kernel by simply shuffling the $(\boldsymbol{u}_i, \boldsymbol{y}_i)$ and $\boldsymbol{v}_i$ across examples. We can also use shuffling to assess whether the optimal value $\hat{\gamma}_{hs}^2$ found during the optimization is significantly different from zero by comparing the value to a null distribution over $\hat{\gamma}_{hs}^2$ obtained from datasets where the $(\boldsymbol{u}_i, \boldsymbol{y}_i)$ and $\boldsymbol{v}_i$ have been permuted across examples.

**Minimization procedure and gradients** For optimizing (6) with respect to $Q$ we use gradient descent over the orthogonal group $SO(n)$. The optimization can be carried out by computing the unconstrained gradient $\nabla_Q \gamma$ of the objective function with respect to $Q$ (treating $Q$ as an ordinary matrix), projecting that gradient onto the tangent space of $SO(n)$, and performing a line search along the gradient direction. We now present the necessary formulae to implement the optimization in a modular fashion. We first show how to compute the gradient $\nabla_Q \gamma$ in terms of the gradients $\nabla_{\boldsymbol{u}_i, \boldsymbol{v}_i} \hat{\gamma}_{hs}^2$, then we show how to compute the $\nabla_{\boldsymbol{u}_i, \boldsymbol{v}_i} \hat{\gamma}_{hs}^2$ in terms of derivatives of kernel functions, and finally demonstrate how the formulae change when approximating the kernel matrices with an incomplete Cholesky decomposition.

Given the unconstrained gradient $\nabla_Q \gamma$ the projection onto the tangent space is given by $\boldsymbol{\zeta} = Q \nabla_Q \gamma^\top Q - \nabla_Q \gamma$ [13, eq. (22)]. The function is then minimized by performing a line-search along $\pi (Q + t\boldsymbol{\zeta})$, where $\pi$ is the projection onto $SO(n)$ which can easily be computed via singular value decomposition of $Q + t\boldsymbol{\zeta}$ and setting the singular values to one [13, prop. 7].

This means that all we need for the gradient descent on $SO(n)$ is the unconstrained gradient $\nabla_Q \gamma$. This gradient takes the form of a sum of outer products [16, eq. (20)]

$$\nabla_Q \hat{\gamma}_{hs}^2 \;\; = \;\; \sum_{i=1}^m \frac{\partial \hat{\gamma}_{hs}^2}{\partial (\boldsymbol{u}_i, \boldsymbol{v}_i)} \cdot \boldsymbol{x}_i^\top = J^\top \Xi, \quad J = \left( \frac{\partial \hat{\gamma}_{hs}^2}{\partial (\boldsymbol{u}_i, \boldsymbol{v}_i)} \right)_i,$$

where the matrix $\Xi$ contains the stimuli $\boldsymbol{x}_i$ in its rows.

The first $k$ columns $J_\eta^{(u)}$ corresponding to the dimension of the features $\boldsymbol{u}_i$ and the last $n-k$ columns $J^{(v)}$ corresponding to the dimension of the features $\boldsymbol{v}_i$ are given by

$$J_\eta^{(u)} = \frac{2}{(m-1)^2} \operatorname{diag}\left( H K_2 H D_\eta^{(u)\top} \right) \quad \text{and} \quad J_\eta^{(v)} = \frac{2}{(m-1)^2} \operatorname{diag}\left( H K_1 H D_\eta^{(v)\top} \right),$$

where

$$\left( D_\eta^{(u)} \right)_{ij} \;\; = \;\; \left( \frac{\partial}{\partial u_{i\eta}} k \left( (\boldsymbol{u}_i, \boldsymbol{v}_i, \boldsymbol{y}_i), (\boldsymbol{u}_j, \boldsymbol{v}_j, \boldsymbol{y}_j) \right) \right)_{ij}$$

contains the partial derivatives of the kernel with respect to the $\eta^{th}$ dimension of $\boldsymbol{u}$ (and analogously for $\boldsymbol{v}$) in the *first* argument (see supplementary material for the derivation).

**Efficient implementation with incomplete Cholesky decomposition of the kernel matrix** So far, the evaluation of HSIC requires the computation of two $m \times m$ kernel matrices in each step. For larger datasets this can quickly become computationally prohibitive. In order to speed up computation time, we approximate the kernel matrices by an incomplete Cholesky decomposition $K = LL^\top$, where $L \in \mathbb{R}^{m \times \ell}$ is a "tall" matrix [1]. In that case, HSIC can be computed much faster as the trace of a product of two $\ell \times \ell$ matrices because

$$\operatorname{tr}\left( K_1 H K_2 H \right) \;\; = \;\; \operatorname{tr}\left( L_1^\top H^2 L_2 L_2^\top H^2 L_1 \right),$$

where $H L_k$ can be efficiently computed by centering $L_k$ on its row mean. Also in this case, the matrix $J$ can be computed efficiently in terms of derivatives of sub-matrices of the kernel matrix (see supplementary material for the exact formulae).

# 3 Related work

**Kernel dimension reduction in regression [5, 6]** Fukumizu and colleagues find maximally informative features $\boldsymbol{U}$ by minimizing $\mathrm{E}_{\boldsymbol{U}} \left[ I \left[ \boldsymbol{V} \mid \boldsymbol{U} : \boldsymbol{Y} \mid \boldsymbol{U} \right] \right]$ in equation (1) via conditional kernel

covariance operators. They show that the covariance operator equals zero if and only if $\boldsymbol{Y}$ is conditionally independent of $\boldsymbol{V}$ given $\boldsymbol{U}$, that is, $\boldsymbol{Y} \perp\!\!\!\perp \boldsymbol{V} \mid \boldsymbol{U}$. In that case, $\boldsymbol{U}$ carries all information about $\boldsymbol{Y}$. Although their approach is closest to ours, it differs in a few key aspects: In contrast to our approach, their objective involves the inversion of a—potentially large—kernel matrix which needs additional regularization in order to be invertible. A conceptual difference is that we are optimizing a slightly more restrictive problem because their objective does not attempt to make $\boldsymbol{U}$ independent of $\boldsymbol{V}$ as well. However, this will not make a difference in many practical cases, since many stimulus distributions are Gaussian for which the dependencies between $\boldsymbol{U}$ and $\boldsymbol{V}$ can be removed by pre-whitening the stimulus data before training LID. In that case $I[\boldsymbol{U} : \boldsymbol{V}] = 0$ for every choice of $Q$ and equation (2) becomes equivalent to maximizing the mutual information between $\boldsymbol{U}$ and $\boldsymbol{Y}$. The advantage of our formulation of the problem is that it allows us to detect and quantify independence by comparing the current $\hat{\gamma}_{hs}$ to its null distribution obtained by shuffling the $(\boldsymbol{y}_i, \boldsymbol{u}_i)$ against $\boldsymbol{v}_i$ across examples. This is hardly possible in the conditional case. Also note that for spherically symmetric data $I[\boldsymbol{U} : \boldsymbol{V}] = \text{const.}$ for every choice of $Q$. In that case equation (2) becomes equivalent to maximizing $I[\boldsymbol{Y} : \boldsymbol{U}]$. However, a residual redundancy remains which would show up when comparing $\hat{\gamma}_{hs}^2$ to its null distribution. Finally, the use of kernel covariance operators is bound to kernels that factorize. In principle, our method is also applicable to non-factorizing kernels if we use $\gamma_{\mathcal{H}}$ instead of $\gamma_{hs}$ and obtain the samples from the product distribution of $P_{\boldsymbol{Y}, \boldsymbol{U}} \times P_{\boldsymbol{V}}$ via shuffling.

**Maximally informative dimensions [22]**    Sharpee and colleagues maximize the relative entropy $I_{\text{spike}} = D_{KL}\left[p\left(\boldsymbol{v}^\top \boldsymbol{s} | \text{spike}\right) \| p\left(\boldsymbol{v}^\top \boldsymbol{s}\right)\right]$ between the distribution of stimuli projected onto informative dimensions given a spike, to the marginal distribution of the projection. This relative entropy is the part of the mutual information which is carried by the arrival of a single spike, since

$$I\left[\boldsymbol{v}^\top \boldsymbol{s} : \{\text{spike, no spike}\}\right] \quad = \quad p\left(\text{spike}\right) \cdot I_{\text{spike}} + p\left(\text{no spike}\right) I_{\text{no spike}}.$$

Their method is also completely non-parametric and captures higher order dependencies between a stimulus and a single spike. However, by focusing on single spikes and the spike triggered density only, it neglects the dependencies between spikes and the information carried by the silence of the neuron [28]. Additionally, the generalization to spike patterns or population responses is non-trivial because the information between the projected stimuli and spike patterns $\boldsymbol{\varpi}_1, ..., \boldsymbol{\varpi}_\ell$ becomes $I\left[\boldsymbol{v}^\top \boldsymbol{s} : \boldsymbol{\varpi}\right] = \sum_i p\left(\boldsymbol{\varpi}_i\right) \cdot I_{\boldsymbol{\varpi}_i}$. This requires the estimation of a conditional distribution $p\left(\boldsymbol{v}^\top \boldsymbol{s} | \boldsymbol{\varpi}_i\right)$ for each pattern $\boldsymbol{\varpi}_i$ which can quickly become prohibitive when the number of patterns grows exponentially.

## 4    Experiments

In all the experiments below, we demonstrate the validity of our methods on controlled artificial examples and on P-unit recordings from electric fish. We use an RBF kernel on the $\boldsymbol{v}_i$ and a tensor RBF kernel on the $(\boldsymbol{u}_i, \boldsymbol{y}_i)$:

$$k\left(\boldsymbol{v}_i, \boldsymbol{v}_j\right) = \exp\left(-\frac{\|\boldsymbol{v}_i - \boldsymbol{v}_j\|^2}{\sigma^2}\right) \quad \text{and} \quad k\left((\boldsymbol{u}_i, \boldsymbol{y}_i), (\boldsymbol{u}_j, \boldsymbol{y}_j)\right) = \exp\left(-\frac{\|\boldsymbol{u}_i \boldsymbol{y}_i^\top - \boldsymbol{u}_j \boldsymbol{y}_j^\top\|^2}{\sigma^2}\right).$$

The derivatives of the kernels can be found in the supplementary material. Unless noted otherwise the $\sigma$ were chosen to be the median of pairwise Euclidean distances between data points. In all artificial experiments, $Q$ was chosen randomly.

**Linear Non-Linear Poisson Model (LNP)**    In this experiment, we trained LID on a simple linear nonlinear Poisson (LNP) neuron $y_i \sim \text{Poisson}\left(\lfloor \langle \boldsymbol{w}, \boldsymbol{x}_i \rangle - \theta \rfloor_+\right)$ with an exponentially decaying filter and a rectifying non-linearity (see Figure 1, left). We used $m = 5000$ data points $\boldsymbol{x}_i$ from a 20-dimensional standard normal distribution $\mathcal{N}(0, I)$ as input. The offset was chosen such that approximately $35\%$ non-zero spike counts in the $y_i$ were obtained. We used one informative and 19 non-informative dimensions, and set $\sigma = 1$ for the tensor kernel.

After optimization, the first dimension $\boldsymbol{q}_1$ of $Q$ converged to the filter $\boldsymbol{w}$ (Figure 1). We compared the HSIC values $\hat{\gamma}_{hs}\left[\{(\boldsymbol{y}_i, \boldsymbol{u}_i)\}_{i=1,...,m} : \{\boldsymbol{v}_i\}_{i=1,...,m}\right]$ before and after the optimization to their null distribution obtained by shuffling. Before the optimization, the dependence of $(\boldsymbol{Y}, \boldsymbol{U})$ and $\boldsymbol{V}$

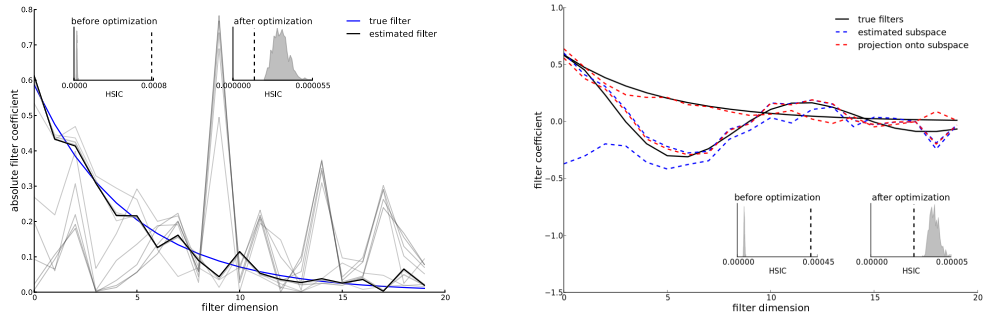

Figure 1: **Left: LNP Model.** The informative dimension (gray during optimization, black after optimization) converges to the true filter of an LNP model (blue line). Before optimization $(\boldsymbol{Y}, \boldsymbol{U})$ and $\boldsymbol{V}$ are dependent as shown by the left inset (null distribution obtained via shuffling in gray, dashed line shows actual HSIC value). After the optimization (right inset) the HSIC value is even below the null distribution. **Right: Two state neuron.** LID correctly identifies the subspace (blue dashed) in which the two true filters (solid black) reside since projections of the filters on the subspace (red dashed) closely resemble the original filters.

is correctly detected (Figure 1, left, insets). After convergence the actual HSIC value lies left to the null distribution's domain. Since the appropriate test for independence would be one-sided, the null hypothesis "$(\boldsymbol{Y}, \boldsymbol{U})$ is independent of $\boldsymbol{V}$" would not be rejected in this case.

**Two state neuron** In this experiment, we simulated a neuron with two states that were both attained in $50\%$ of the trials (see Figure 1, right). This time, the output consisted of four "bins" whose statistics varied depending on the state. In the first—steady rate—state, the four bins contained spike counts drawn from an LNP neuron with exponentially decaying filter as above. In the second—burst—state, the first two bins were drawn from Poisson distribution with a fixed base rate independent of the stimulus. The second two bins were drawn from an LNP neuron with a modulated exponential filter and higher gain. We used $m = 8000$ input stimuli from a 20-dimensional standard normal distribution. We use two informative dimensions and set $\sigma$ of the tensor kernel to two times the median of the pairwise distances. LID correctly identified the subspace associated with the two filters also in this case (Figure 1, right).

**Artificial complex cell** In a second experiment, we estimated the two-dimensional subspace associated with a artificial complex cell. We generated a quadrature pair $\boldsymbol{w}_1$ and $\boldsymbol{w}_2$ of two 10-dimensional filters (see Figure 2, left). We used $m = 8000$ input points from a standard normal distribution. Responses were generated from a Poisson distribution with the rate given by $\lambda_i = \langle \boldsymbol{w}_1, \boldsymbol{x}_i \rangle^2 + \langle \boldsymbol{w}_2, \boldsymbol{x}_i \rangle^2$. This led to about $34\%$ non-zero neural responses. When using two informative subspaces, LID was able to identify the subspace correctly (Figure 2, left). When comparing the HSIC value against the null distribution found via shuffling, the final value indicated no further dependencies. When only a one-dimensional subspace was used (Figure 2, right), LID did not converge to the correct subspace. Importantly, the HSIC value after optimization was clearly outside the support of the null distribution, thereby correctly indicating residual dependencies.

**P-Unit recordings from weakly electric fish** Finally, we applied our method to P-unit recordings from the weakly electric fish *Eigenmannia virescens*. These weakly electric fish generate a dipole-like electric field which changes polarity with a frequency at about 300Hz. Sensors in the skin of the fish are tuned to this carrier frequency and respond to amplitude changes caused by close-by objects with different conductive properties than water [20]. In the present recordings, the immobilized fish was stimulated with 10s of $300 - 600$Hz low-pass filtered full field frozen Gaussian white noise amplitude modulations of its own field. Neural activity was recorded intra-cellularly from the P-unit afferents.

Spikes were binned with 1ms precision. We selected $m = 8400$ random time points in the spike response and the corresponding preceding 20ms of the input (20 dimensions). We used the same

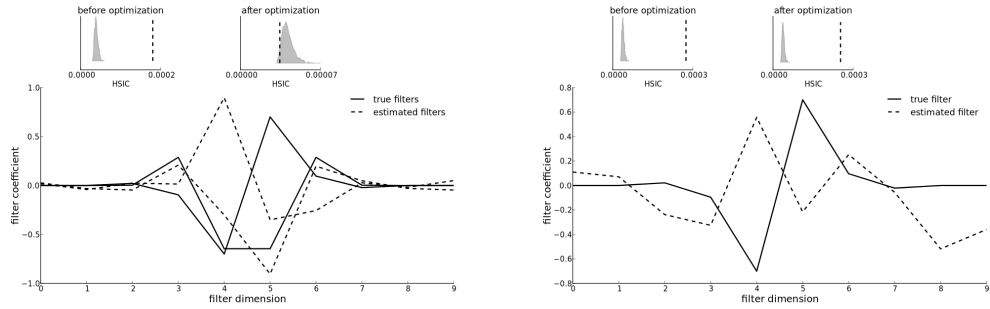

Figure 2: **Artificial Complex Cell. Left:** The original filters are 90° phase shifted Gabor filters which form an orthogonal basis for a two-dimensional subspace. After optimization, the two informative dimensions of LID (first two rows of $Q$) converge to that subspace and also form a pair of 90° phase shifted filters (note that even if the filters are not the same, they span the same subspace). Comparing the HSIC values before and after optimization shows that this subspace contains the relevant information (left and right inset). **Right:** If only a one-dimensional informative subspace is used, the filter only slightly converges to the subspace. After optimization, a comparison of the HSIC value to the null distribution obtained via shuffling indicates residual dependencies which are not explained by the one-dimensional subspace (left and right inset).

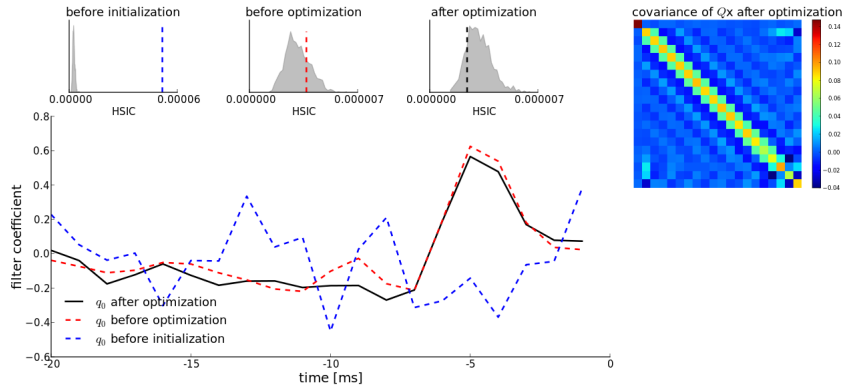

Figure 3: **Most informative feature for a weakly electric fish P-Unit:** A random filter (blue trace) exhibits HSIC values that are clearly outside the domain of the null distribution (left inset). Using the spike triggered average (red trace) moves the HSIC values of the first feature of $Q$ already inside the null distribution (middle inset). Further optimization with LID refines the feature (black trace) and brings the HSIC values closer to zero (right inset). After optimization, the informative feature $U$ is independent of the features $V$ because the first row and column of the covariance matrix of the transformed Gaussian input show no correlations. The fact that one informative feature is sufficient to bring the HSIC values inside the null distribution indicates that a single subspace captures all information conveyed by these sensory neurons.

kernels as in the experiment on the LNP model. We initialized the first row in $Q$ with the normalized spike triggered average (STA; Figure 3, left, red trace). We neither pre-whitened the data for computing the STA nor for the optimization of LID. Unlike a random feature (Figure 3, left, blue trace), the spike triggered average already achieves HSIC values within the null distribution (Figure 3, left and middle inset). The most informative feature corresponding to $U$ looks very similar to the STA but shifts the HSIC value deeper into the domain of the null distribution (Figure 3, right inset).

This indicates that one single subspace in the input is sufficient to carry all information between the input and the neural response.

## 5 Discussion

Here we presented a non-parametric method to estimate a subspace of the stimulus space that contains all information about a response variable $Y$. Even though our method is completely generic and applicable to arbitrary input-output pairs of data, we focused on the application in the context of sensory neuroscience. The advantage of the generic approach is that $Y$ can in principle be anything from spike counts, to spike patterns or population responses. Since our method finds the most informative dimensions by making the complement of those dimensions as independent from the data as possible, we termed it *least informative dimensions (LID)*. We use the Hilbert-Schmidt independence criterion to minimize the dependencies between the uninformative features and the combination of informative features and outputs. This measure is easy to implement, avoids the need to estimate mutual information, and its estimator has good convergence properties independent of the dimensionality of the data. Even though our approach only estimates the informative features and not mutual information itself, it can help to estimate mutual information by reducing the number of dimensions.

As in the approach by Fukumizu and colleagues, it might be that no $Q$ exists such that $I[Y, U : V] = 0$. In that situation, the price to pay for an easier measure is that it is hard to make definite statements about the informativeness of the features $U$ in terms of the Shannon information, since $\gamma_{\mathcal{H}} = I[Y, U : V] = 0$ is the point that connects $\gamma_{\mathcal{H}}$ to the mutual information. As demonstrated in the experiments, we can detect this case by comparing the actual value of $\hat{\gamma}_{\mathcal{H}}$ to an empirical null distribution of $\hat{\gamma}_{\mathcal{H}}$ values obtained by shuffling the $v_i$ against the $u_i, y_i$ pairs. However, if $\gamma_{\mathcal{H}} \neq 0$, theoretical upper bounds on the mutual information are unfortunately not available. In fact, using results from [25] and Pinsker's inequality one can show that $\gamma_{\mathcal{H}}^2$ bounds the mutual information from *below*. One might now be tempted to think that maximizing $\gamma_{\mathcal{H}}[Y, U]$ might be a better way to find informative features. While this might be a way to get some informative features [24], it is not possible to link the features to informativeness in terms of *Shannon* mutual information, because the point that builds the bridge between the two dependency measures is where both of them are zero. Anywhere else the bound may not be tight so the maximally informative features in terms of $\gamma_{\mathcal{H}}$ and in terms of mutual information can be different.

Another problem our approach shares with many algorithms that detect higher-order dependencies is the non-convexity of the objective function. In practice, we found that the degree to which this poses a problem very much depends on the problem at hand. For instance, while the subspaces of the LNP or the two state neuron were detected reliably, the two dimensional subspace of the artificial complex cell seems to pose a harder problem. It is likely that the choice of kernel has an influence on the landscape of the objective function. We plan to explore this relationship in more detail in the future. In general, a good initialization of $Q$ helps to get close to the global optimum.

Beyond that, however, integral probability metric approaches to maximally informative dimensions offer a great chance to avoid many problems associated with direct estimation of mutual information, and to extend it to much more interesting output structures than single spikes.

## Acknowledgements

Fabian Sinz would like to thank Lucas Theis and Sebastian Gerwinn for helpful discussions and comments on the manuscript. This study is part of the research program of the Bernstein Center for Computational Neuroscience, Tübingen, funded by the German Federal Ministry of Education and Research (BMBF; FKZ: 01GQ1002).

## Footnotes

[1] With some abuse of notation, we wrote MMD as a function of the probability measures.

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
