[Supplementary Material]

# 1 Supplementary Material

## 1.1 Derivation of equation (2)

The multi-information is defined as

$$I\left[\boldsymbol{Y}:\boldsymbol{U}:\boldsymbol{V}\right] \;=\; \left\langle \log \frac{p\left(\boldsymbol{y},\boldsymbol{u},\boldsymbol{v}\right)}{p\left(\boldsymbol{y}\right)p\left(\boldsymbol{u}\right)p\left(\boldsymbol{v}\right)} \right\rangle_{\boldsymbol{Y},\boldsymbol{U},\boldsymbol{V}}.$$

It satifies the chain rule

$$I\left[\boldsymbol{Y}:\boldsymbol{U}:\boldsymbol{V}\right] \;=\; I\left[(\boldsymbol{Y},\boldsymbol{U}):\boldsymbol{V}\right]+I\left[\boldsymbol{Y}:\boldsymbol{U}\right].$$

Therefore,

$$
\begin{aligned}
I\left[\boldsymbol{Y}:\boldsymbol{U}:\boldsymbol{V}\right] &= I\left[(\boldsymbol{Y},\boldsymbol{U}):\boldsymbol{V}\right]+I\left[\boldsymbol{Y}:\boldsymbol{U}\right] \\
&= I\left[\boldsymbol{Y}:(\boldsymbol{U},\boldsymbol{V})\right]+I\left[\boldsymbol{U}:\boldsymbol{V}\right] \\
\Leftrightarrow I\left[(\boldsymbol{Y},\boldsymbol{U}):\boldsymbol{V}\right] &= I\left[\boldsymbol{Y}:(\boldsymbol{U},\boldsymbol{V})\right]+I\left[\boldsymbol{U}:\boldsymbol{V}\right]-I\left[\boldsymbol{Y}:\boldsymbol{U}\right] \\
&= I\left[\boldsymbol{Y}:\boldsymbol{X}\right]+I\left[\boldsymbol{U}:\boldsymbol{V}\right]-I\left[\boldsymbol{Y}:\boldsymbol{U}\right].
\end{aligned}
$$

## 1.2 Kernels and their derivatives

**RBF kernel**  The RBF kernel is given by

$$k\left(\boldsymbol{x}_i,\boldsymbol{x}_j\right) \;=\; \exp\left(-\frac{\|\boldsymbol{x}_i-\boldsymbol{x}_j\|^2}{\sigma^2}\right).$$

Its derivative w.r.t. $\boldsymbol{x}_i$ is

$$\frac{\partial}{\partial \boldsymbol{x}_i}k\left(\boldsymbol{x}_i,\boldsymbol{x}_j\right) \;=\; k\left(\boldsymbol{x}_i,\boldsymbol{x}_j\right)\cdot -\frac{2}{\sigma^2}\left(\boldsymbol{x}_i-\boldsymbol{x}_j\right).$$

**RBF tensor kernel**  The RBF tensor kernel is given by

$$
\begin{aligned}
k\left((\boldsymbol{x}_1,\boldsymbol{y}_1),(\boldsymbol{x}_2,\boldsymbol{y}_2)\right) &= \exp\left(-\frac{\|\boldsymbol{x}_1\otimes\boldsymbol{y}_1-\boldsymbol{x}_2\otimes\boldsymbol{y}_2\|_2^2}{\sigma^2}\right) \\
\|\boldsymbol{x}_1\otimes\boldsymbol{y}_1-\boldsymbol{x}_2\otimes\boldsymbol{y}_2\|_2^2 &= \langle\boldsymbol{x}_1\otimes\boldsymbol{y}_1,\boldsymbol{x}_1\otimes\boldsymbol{y}_1\rangle-2\langle\boldsymbol{x}_1\otimes\boldsymbol{y}_1,\boldsymbol{x}_2\otimes\boldsymbol{y}_2\rangle+\langle\boldsymbol{x}_2\otimes\boldsymbol{y}_2,\boldsymbol{x}_2\otimes\boldsymbol{y}_2\rangle \\
&= \langle\boldsymbol{x}_1,\boldsymbol{x}_1\rangle\langle\boldsymbol{y}_1,\boldsymbol{y}_1\rangle-2\langle\boldsymbol{x}_1,\boldsymbol{x}_2\rangle\langle\boldsymbol{y}_1,\boldsymbol{y}_2\rangle+\langle\boldsymbol{x}_2,\boldsymbol{x}_2\rangle\langle\boldsymbol{y}_2,\boldsymbol{y}_2\rangle.
\end{aligned}
$$

The derivative of $k$ w.r.t. $\boldsymbol{x}_1$ and $\boldsymbol{y}_2$ are given by

$$
\begin{aligned}
\frac{\partial}{\partial \boldsymbol{x}_1}k\left((\boldsymbol{x}_1,\boldsymbol{y}_1),(\boldsymbol{x}_2,\boldsymbol{y}_2)\right) &= k\left((\boldsymbol{x}_1,\boldsymbol{y}_1),(\boldsymbol{x}_2,\boldsymbol{y}_2)\right)\cdot -\frac{2}{\sigma^2}\left(\langle\boldsymbol{y}_1,\boldsymbol{y}_1\rangle\boldsymbol{x}_1-\langle\boldsymbol{y}_1,\boldsymbol{y}_2\rangle\boldsymbol{x}_2\right) \\
\frac{\partial}{\partial \boldsymbol{y}_1}k\left((\boldsymbol{x}_1,\boldsymbol{y}_1),(\boldsymbol{x}_2,\boldsymbol{y}_2)\right) &= k\left((\boldsymbol{x}_1,\boldsymbol{y}_1),(\boldsymbol{x}_2,\boldsymbol{y}_2)\right)\cdot -\frac{2}{\sigma^2}\left(\langle\boldsymbol{x}_1,\boldsymbol{x}_1\rangle\boldsymbol{y}_1-\langle\boldsymbol{x}_1,\boldsymbol{x}_2\rangle\boldsymbol{y}_2\right).
\end{aligned}
$$

## 1.3 Computation of $J$

**For the regular case**  For HSIC, the matrix $J$ can be computed in terms of the partial derivatives

$$\left(D_\eta^{(u)}\right)_{ij} \;=\; \left(\frac{\partial}{\partial u_{i\eta}}k\left((\boldsymbol{u}_i,\boldsymbol{v}_i,\boldsymbol{y}_i),(\boldsymbol{u}_j,\boldsymbol{v}_j,\boldsymbol{y}_j)\right)\right)_{ij}$$

of the kernel with respect to the $\eta^{th}$ dimension of $\boldsymbol{u}$ (and analogously for $\boldsymbol{v}$) in the *first* argument, even if $i=j$.

In general, consider any function $f$ that depends on a kernel matrix $K$ which in turn depends on set of data points $\boldsymbol{u}_i$ collected in the rows of a matrix $\Upsilon$. Since $K_{ij}$ only depends on the $i^{th}$ and $j^{th}$ example, the derivative $\frac{\partial f}{\partial u_{\nu\eta}}$ can be written as

$$\frac{\partial f}{\partial u_{\nu\eta}} = \sum_{i,j=1}^m \left(\frac{\mathrm{d}f}{\mathrm{d}K}\right)_{ij}\frac{\mathrm{d}k_{ij}}{\mathrm{d}u_{\nu\eta}}\left(\delta_{i\nu}+\delta_{j\nu}\right) \quad \text{or} \quad \frac{\partial f}{\partial \Upsilon_{:\eta}} = \mathrm{diag}\left(\left(\frac{\partial f}{\partial K}+\frac{\partial f}{\partial K}^\top\right)D_\eta^{(u)\top}\right), \tag{1.1}$$

where $\Upsilon_{:\eta}$ denotes the $\eta$th column of $\Upsilon$. With $f = \mathrm{tr}$ and $\frac{\partial}{\partial K_1}\mathrm{tr}\,(K_1 H K_2 H) = H K_2 H$, the derivatives in $J = \left(J^{(u)}, J^{(v)}\right)$ can be generically computed as a function of the derivatives of kernels $D_\eta^{(u)}$ and $D_\eta^{(v)}$:

$$
\begin{aligned}
J_\eta^{(u)} &= \frac{2}{(m-1)^2}\mathrm{diag}\left(H K_2 H D_\eta^{(u)\top}\right) \\
J_\eta^{(v)} &= \frac{2}{(m-1)^2}\mathrm{diag}\left(H K_1 H D_\eta^{(v)\top}\right),
\end{aligned}
$$

since $\mathrm{tr}\,(K_1 H K_2 H) = \mathrm{tr}\,(K_2 H K_1 H)$.

**For the incomplete Cholesky decomposition**  When computing the derivative of $\hat{\gamma}_{hs}^2$ with the incomplete Cholesky decomposition, we need to take into account that (i) each entry in the kernel matrix might now be a function of more than a pair of data points, and we (ii) want to avoid having to compute the whole kernel matrix. In order to compute the derivative note that the approximation $\tilde{K} = LL^\top$ to $K$ is given by

$$
K \approx \tilde{K} = LL^\top = \left(\begin{array}{cc} K_{ii} & K_{ij} \\ K_{ii}^\top & K_{ij}^\top K_{ii}^{-1} K_{ij} \end{array}\right),
$$

where $i$ is an index set containing the indices of the pivot elements used to compute the incomplete Cholesky decomposition and $j = \{1, ..., m\} \setminus i$ is its complement [1]. Therefore,

$$
\mathrm{tr}\left(\tilde{K}\underbrace{H\tilde{K}_2 H}_{=: A^{(2)}}\right) = \mathrm{tr}\left(K_{ii} A_{ii}^{(2)}\right) + \mathrm{tr}\left(K_{ij} A_{ji}^{(2)}\right) + \mathrm{tr}\left(K_{ji} A_{ij}^{(2)}\right) + \mathrm{tr}\left(K_{ij}^\top K_{ii}^{-1} K_{ij} A_{jj}^{(2)}\right),
$$

where indexing with the index sets $i$ and $j$ denotes the extraction of a sub-matrix of the respective matrix.

We can now take the derivatives of $\hat{\gamma}_{hs}^2$ with respect to the pivot and non-pivot elements (corresponding to the index sets $i$ and $j$ and—equivalently—to rows of $J$). Note that equation (1.1) becomes $\frac{\partial f}{\partial \Upsilon_{:\eta}} = \mathrm{diag}\left(\frac{\partial f}{\partial K} D_\eta^\top\right)$ in the case of the cross-kernel matrix $K_{ij}$. Using the product rule for matrix derivatives [2], this reduces the derivative of the approximate case to the one above since

$$
\begin{aligned}
\frac{\partial}{\partial K_{ji}}\mathrm{tr}\left(K_{ij}^\top K_{ii}^{-1} K_{ij} A_{jj}^{(2)}\right) &= K_{ii}^{-1} K_{ij} A_{jj}^{(2)} + \left(A_{jj}^{(2)} K_{ji} K_{ii}^{-1}\right)^\top \\
\frac{\partial}{\partial K_{ii}}\mathrm{tr}\left(K_{ji} K_{ii}^{-1} K_{ij} A_{jj}^{(2)}\right) &= -K_{ii}^{-1} K_{ij} A_{jj}^{(2)} K_{ji} K_{ii}^{-1}.
\end{aligned}
$$

Let $K := K_1$, $A^{(2)} := H\tilde{K}_2 K$, and $i$ and $j$ the pivot and non-pivot indices of $K_1$. Then the first $k$ columns (corresponding to the features $u_i$) of $J$ are given by

$$
\begin{aligned}
J_{i\eta}^{(u)} &= \frac{2}{(m-1)^2}\left(\mathrm{diag}\left(A_{ii}^{(2)} D_{ii\eta}^{(u)\top}\right) + \mathrm{diag}\left(A_{ij}^{(2)} D_{ij\eta}^{(u)\top}\right) + \mathrm{diag}\left(K_{ii}^{-1} K_{ij} A_{jj}^{(2)} D_{ij\eta}^{(u)\top}\right) - \mathrm{diag}\left(K_{ii}^{-1} K_{ij} A_{jj}^{(2)} K_{ij}^\top K_{ii}^{-1} D_{ii\eta}^{(u)\top}\right)\right) \\
J_{j\eta}^{(u)} &= \frac{2}{(m-1)^2}\left(\mathrm{diag}\left(A_{ji}^{(2)} D_{ji\eta}^{(u)\top}\right) + \mathrm{diag}\left(A_{jj}^{(2)} K_{ij}^\top K_{ii}^{-1} D_{ji\eta}^{(u)\top}\right)\right).
\end{aligned}
$$

Let $K := K_2$, $A^{(1)} := H\tilde{K}_1 H$, and $i$ and $j$ the pivot and non-pivot indices of $K_2$. Then the last $n - k$ columns (corresponding to the features $v_i$) of $J$ are given by

$$
\begin{aligned}
J_{i\eta}^{(v)} &= \frac{2}{(m-1)^2}\left(\mathrm{diag}\left(A_{ii}^{(1)} D_{ii\eta}^{(v)\top}\right) + \mathrm{diag}\left(A_{ij}^{(1)} D_{ij\eta}^{(v)\top}\right) + \mathrm{diag}\left(K_{ii}^{-1} K_{ij} A_{jj}^{(1)} D_{ij\eta}^{(v)\top}\right) - \mathrm{diag}\left(K_{ii}^{-1} K_{ij} A_{jj}^{(1)} K_{ij}^\top K_{ii}^{-1} D_{ii\eta}^{(v)\top}\right)\right) \\
J_{j\eta}^{(v)} &= \frac{2}{(m-1)^2}\left(\mathrm{diag}\left(A_{ji}^{(1)} D_{ji\eta}^{(v)\top}\right) + \mathrm{diag}\left(A_{jj}^{(1)} K_{ij}^\top K_{ii}^{-1} D_{ji\eta}^{(v)\top}\right)\right).
\end{aligned}
$$