[Reviews · NeurIPS 2013]

Submitted by Assigned_Reviewer_5

Review for # 271: “Least Informative Dimensions”

This paper addresses the problem of finding the low dimensional stimulus subspaces (of a high dimensional stimulus space) that most influence the spiking of a neuron or potentially other process. In contrast to Maximally informative dimensions, which directly looks for informative subspaces, this method tries to decompose the stimulus space into two orthogonal subspaces by making one of them as uninformative about the spiking as possible. The methodology is highly advanced, involving replacing the relative entropy by an approximate “integral probability metric” and representing this metric in terms of a kernel function. After making some simplifying assumptions regarding the form of this kernel, the authors use a gradient ascent. They give detailed formulas for the required derivatives in the supplement. They then apply their method to several simulated data sets and to a neuron recorded in the electric fish. Using a shuffling procedure to calculate confidence bound upon the integral probability metric they show that for their examples, this procedure finds a subspace of the correct dimension which contains all the relevant information in the stimulus.

This paper was somewhat dense, but I enjoyed it once I figured out what was going on. The problem of finding informative subspaces is an important one, and new methods are welcome. The math is rigorous and the examples in the results section, while somewhat simple, are appropriate for a NIPS paper given the space constraints.

One thing that is missing is any mention of computation times and a comparison with more standard methods such as Sharpee’s maximally informative dimensions. Given that the authors are motivating their work by the computational intensity of directly estimating the mutual information, I think that they should compare the two methods both in their computation time, and also their accuracy.

A second question I had regarded the firing rates used in their simulated examples. They state on line 257 that “35% non-zero spike counts were obtained” for the LNP neuron. This seems rather high, as in 350 Hz at 1ms resolution and 35 Hz at 10 ms resolution. Can the method detect the relevant dimension (particularly in the complex cell example) at lower resolution?

In summary I think this is a nice paper suitable for NIPS if the above concerns are addressed.
Summary: A new method for finding stimulus subspaces which are informative about neural spiking (or other random processes). Very mathematically rigorous and quite interesting. However, no direct comparison with other methods (such as maximally informative dimensions) are provided and given the claim of efficiency (or at least the motivation of efficiency) details such as computation times etc. should be included.

Submitted by Assigned_Reviewer_6

This paper is devoted to the problem of finding relevant stimulus features from the neural responses. The main idea is to minimize the information between a certain set of features and the neural responses combined with the remaining stimulus features in order to separate the relevant from irrelevant features. The method builds on the prior work by Fukumizu and colleagues, but unfortunately makes an additional restriction that greatly diminishes the potential use of this method. Specifically, the irrelevant and relevant dimensions need to be statistically independent from each other within the stimulus ensemble. While the authors argue that this is not a crucial assumption “because many distributions are Gaussian for which the dependencies between U [informative features] and V[the uninformative features] can be removed by pre-whitening the stimulus data before training”, this assumption essentially eliminates the novel contribution from this method. This is because with correlated Gaussian stimuli one can use spike-triggered covariance (STC) to find all features that are informative about the neural response in one simple computational step that avoids the non-convexity and complexity of the proposed optimization. Furthermore, the STC method has recent been shown to apply to elliptically symmetric stimuli (Samengo & Gollisch J Computational Neuroscience 2013). Finally, the proposed method has to iteratively increase the number of relevant features, whereas this is not needed in the STC method.

In terms of practical implementations, the method was only demonstrated in this paper on white noise signals. Even in this simple case, the performance exhibited substantial biases. For example, in Figure 1, right panel, a substantial component along the irrelevant subspace remained after optimization. Note that it is not correct to claim that (lines 286-888, and also lines 315-316) that “LID correctly identifies the subspace (blue dashed) in which the two true filters (solid black) reside since projections of the filters on the subspace closely resemble the original filters. “ The point of this method is to *eliminate* irrelevant dimensions, and this was not achieved.

The superior convergence properties of the proposed techniques relatively to spike triggered average/covariance were not demonstrated. In fact, the method did not even remove biases in the STA, which would be possible using the linear approximation (Figure 3). Note that the Figure 3 description was contradictory: stimuli were described as white noise, yet the footnote stated that the data were not whitened.

The use of information minimization for recovering multiple features and correlated stimuli was demonstrated in Fitzgerald et al. PLoS Computational Biology 2011, “Second-order dimensionality reduction using minimum and maximum mutual information models.”

Line 234-235: it is not correct that MID (also STC) “requires data that contains several spike responses to the same stimulus” – one can optimize the relative entropy based on a sequence of stimuli presented only once in either MID/STC methods.

Lines 236-237: MID/STC can be used with spike patterns by defining events across time bins or different neurons.

The abstract was written in a somewhat misleading manner, because it is not the information that is being minimized, but a related a quantity in combination with crucial assumptions that are not spelled out in the abstract. For example, that kernels need to factorize. The last statement in the abstract is also not clear “… if we can make the uninformative features independent of the rest.” Here, it is not clear that this implies that inputs are essentially required to be uncorrelated if any of the input features can be informative for one of the neurons in a dataset.
Summary: This manuscript describes a method for finding multiple features in neuroscience dimensions. The paper is written fairly clearly. Unfortunately, this method implementation leaves much to be desired compared to the existing methods (e.g. STC) and does not offer new any capabilities because it requires uncorrelated inputs with which STC can be used.

Submitted by Assigned_Reviewer_7

The gist of this paper is essentially a non-parametric method that can be
used to estimate Maximally Informative Dimensions (MID; due to Sharpee et
al). MID is one of the only spike-triggered characterization techniques
that is capable of extracting multiple dimensions that are informative with
respect to higher-order statistics (unlike classic methods such as STA/STC
which rely on changes in mean and variance, respectively). Extending MID to
multiple dimensions is also difficult due to repeated estimation of
relative entropy which quickly becomes intractable due to the amount of
data required as dimensionality increases. Here, the authors use an
'indirect' estimation approach, whereby they can estimate maximally
informative features without actually having to estimate mutual
information.

This paper is largely excellent. The methodology itself is similar in flavour to
the work of Fukumizu et al, but the similarities and conceptual differences
are discussed in detail by the authors. My main criticism is that the paper
is very methods heavy, with a slightly less focus on experiments. This
really does seem like a great estimation technique, and I do not feel that
the experiments sections does it justice. Ideally, I would like to have
seen a detailed comparison between MID and LID in real data (perhaps visual
recordings, similar to the data commonly presented in papers by the Sharpee
group). I appreciate that the authors did present some real data
application in the form of P-unit recordings from weakly electric fish,
although one might argue that this data is non standard in the sense that
the vast majority of spike-triggered characterization techniques are
applied to either the visual or auditory domains. The most disappointing
aspects of the results however, is that the authors do not seem to address
the issue of taking MID to high dimensions - which is one of the key areas
in which current MID estimation algorithms struggle. Some examples of this
can be found in the recent papers by Atencio, Sharpee, and Schreiner, where
the MID approach is applied to the auditory system. An approach like this
could benefit these kind of studies greatly, giving them the ability to
investigate higher-dimensional feature spaces.
A final comment is one of computational efficiency - given that such an
approach is likely to be applied to vast amounts of neural data, it would
be nice to get a feel for how long the algorithm takes to run for data sets
of different sizes.

The paper itself is very well written and particularly clear. It does not
seem to suffer from any obvious grammatical or mathematical errors (at
least, to my eyes).

I very much enjoyed this paper. It is a great example of what a good NIPS
paper should be - it uses modern machine learning methods to study problems
in neuroscience. Spike-triggered characterization methods have been used
for decades, and yet the seemingly simple problem of dealing with
high-dimensional feature spaces has been plagued with difficulties. This
paper provides steps towards solving this problem, which will have long
lasting implications in both neuroscience and machine learning communities.
The only negative aspect of this paper, as I mentioned earlier, is that the
results section is somewhat lacking, and does not fully reflect how good
the estimation technique seems to be.
Summary: An excellent methodology paper, only slightly lacking in an adequate display of results and relevant comparisons to related methods.

Submitted by Assigned_Reviewer_8

I think this paper has some significant strengths - I certainly rank it higher than a 2. There are some nice ideas; it's a well-written paper, and I enjoyed reading it. But there are some significant weaknesses here that need to be addressed.

The basic question that is not addressed sufficiently directly here: is the proposed method actually better than MID or the method of Fukumizu et al, either in terms of computational efficiency or statistical accuracy? (Presumably not the latter, since MID is asymptotically efficient.) Direct comparisons need to be provided. In addition, the numerical examples provided here are rather small.  How well do the methods scale? The authors really need to do a better job making the point that the proposed methods offer something that improves on the state of the art.

In addition, a basic weakness of the proposed method should be addressed more clearly. Specifically, on L77, "if Q can be chosen such that I [Y ;U : V ] = 0" - as noted by reviewer 2, this typically won't happen in the case of naturalistic stimuli X (unlike the case of gaussian X, where we can easily prewhiten), where these information-theoretic methods are of most interest. So this makes the proposed methods seem significantly less attractive. This is a key point here, and needs to be addressed more effectively.


Minor comments -

L81 - "another dependency measure which ... shares its minimum with mutual information, that is, it is zero if and only if the mutual information is zero." - these two statements don't appear to be equivalent in general.  if there is no Q for which the corresponding MI is zero, then it's not clear that these two dependency measures will in fact share the same (arg) min.  this point should be clarified here.

Maybe worth writing out a brief derivation for eq (2), at least in the supplement.

It would be nice to have a clearer description of how the incomplete cholesky decomposition helps - at least mention what the computational complexity is.  The kernel ICA paper by Bach and Jordan does a good job of laying out these issues. Note that we still have to recompute the incomplete cholesky for each new Q (if I understand correctly).

The relative drawbacks of MID seem overstated.  While Sharpee's implementation of MID bins the projected data, there's no conceptual need to do this - a kernel (unbinned) estimator for the required densities could be used instead (and would probably make the gradients easier to compute), or other estimators for the information could be used that don't require density estimates at all.  Similarly, as one of the other reviewers notes, the MID method could easily be applied to multi-spike patterns.  I also agree with the second reviewer that L235 is incorrect and needs revision.
Summary: I think this paper has some significant strengths. There are some nice ideas; it's a well-written paper, and I enjoyed reading it. But there are some significant weaknesses here that need to be addressed.
Author Feedback

Author rebuttal: We thank all reviewers for their constructive feedback. We will make an effort to address all your minor comments in a revised version of the paper. Here, we hope to clarify the major points.

First of all, we would like to stress that our primary motivation was not to improve the runtime of MID but to offer a generalization that extends to populations and spike patterns (we explain below why we think this is difficult with MID). Given the page constraint, we felt that it is more important to demonstrate our method on controlled examples of increasing complexity than to compare it to other algorithms. Furthermore, we are unsure how meaningful a runtime comparison would be, since LID solves a more general problem than MID. We agree that a more extensive experimental exploration is the next step after demonstrating the effectiveness of the method on controlled examples. We will do that in the near future. In the final version of the paper, we will adapt our discussion of MID to avoid giving the impression our goal was to improve the runtime.

Concerning the generality of our method, reviewer #6(2) claimed that it would not yield any novel contribution beyond STC for Gaussian stimuli since STC were sufficient to extract all informative features for correlated Gaussian inputs already. This is not the case for two reasons:

1. STC and STA characterize the spike triggered ensemble which essentially captures the informative features for one single bin. In contrast, our method captures informative stimulus features about more complex responses, including patterns of spikes or population responses. Therefore, our method solves a more general problem than STC and STA independent of the stimulus distribution.

2. STC does not necessarily identify all informative features, even for Gaussian stimuli. This is because the spike triggered distribution does not need to be Gaussian anymore as this simple example illustrates: Let x in R^2 and ~N(0,I), and let y=1 if a<=|x_1|<=b and 0 otherwise, where a and b are chosen such that var(x_1)=1 (this can easily be done using the cdf and ppf of the Gaussian). The spike triggered distribution p(x|y=1) is still white and has mean zero. Therefore, neither STA nor STC can detect the dependencies between stimulus and response. Our algorithm, on the other hand, detects the correct subspace [1,0]'. A similar example can be constructed for correlated Gaussian stimuli. Therefore, our algorithm is more general than STA and STC, even for Gaussian stimuli.

Reviewer #6(2) also raised the concern that a substantial component of the irrelevant dimensions remains after optimization (Fig 1 right panel). Here, we would like to stress that LID identifies the informative _subspace_. Thus, the resulting basis vectors can be flipped or linear combinations of the original filters. If the original filters are not orthogonal, the subspace basis will look different from the filters (since Q is in SO(n)). If the subspace is determined correctly, however, it should contain the original filters. This is indeed the case for Fig. 1/right because substantial parts of the projected filters would be missing otherwise. We will make this point clearer in a final version.

Our reasoning why an extension of MID to patterns and populations is not straightforward is the following: MID maximizes the information between a _single_ spike and the projection of the stimuli onto a particular subspace (which is not equal to I[Y:v'*X], see related work section in the paper). Assuming that the p(v'x|no spike) is very similar to the prior distribution p(v'x), the information of a single spike is approximately the information carried by a single bin. As the single bins can be correlated, it may be desirable to extend MID to several bins. When using spike patterns or population responses, however, there are several pattern triggered ensembles p(v'x|pattern 1), p(v'x|pattern 2), ... which are unequal to the prior distribution p(v'x) and, therefore, carry information (see the equation in line 232 in the related work section). In that case, I[Y:v'*X] has a term for each of those patterns and MID needs to estimate the information for each of them in order to account for the full stimulus response information. Depending on the number of patterns, this can become substantially more involved.

Concerning the independence between U and V, and the choice of Q such that I[Y,U:V]=0, we would like to emphasize that we obtain several advantages by phrasing the objective as I[Y,U:V]= I[Y:X] + I[U:V] − I[Y:U] while maintaining applicability to most common stimulus distributions because I[U:V]=const for most of them. This means that minimizing I[Y,U:V] becomes equivalent to maximizing I[Y:U]. Clearly, for white and correlated Gaussians, I[U:V]=0 after pre-whitening. For elliptically symmetric distributions, pre-whitening yields a spherically symmetric distribution which means that I[U:V]=const, since U and V correspond to coordinates in an orthogonal basis. Since natural signals like images and sound are well described by elliptically symmetric distributions (see work by Lyu/Simoncelli and Hosseini/Sinz/Bethge) I[U:V]=const as well. This also means that even if Q cannot be chosen such that I[Y,U:V]=0, the objective function is still sensible. The only reason why we need I[Y,U:V]=0 is to tie the minimum of the integral probability metric (IPM) to the minimum of the Shannon information. However, even if that minimum cannot be attained, the IPM objective will still yield reasonable features.

In return for the additional restriction and the use of the IPM we get an algorithm that (i) naturally extends to patterns and populations while maintaining computational feasibility, (ii) can assess the Null distribution via permutation analysis (as opposed to Fukumizu et al.), and (iii) does not have to use factorizing joint kernels (as opposed to Fukumizi et al.; we only chose factorizing kernels for convenience).